# Serum lactate dehydrogenase is associated with impaired lung function: NHANES 2011–2012

**Sheng Hu[☯], Jiayue Ye[☯], Qiang Guo[☯], Sheng Zou, Wenxiong Zhang, Deyuan Zhang, Yang Zhang, Silin Wang, Lang Su, Yiping Wei[ID]\***

Department of Thoracic Surgery, The Second Affiliated Hospital of Nanchang University, Nanchang, Jiangxi Province, P. R. China

☯ These authors contributed equally to this work.
\* ndefy08025@ncu.edu.cn

**Data Availability Statement:** All relevant data are within the paper and its Supporting Information files.

## Abstract

### Background

Serum lactate dehydrogenase levels reflect disease status in a variety of organs, but its role in indicating pulmonary function is not yet clear. Therefore, this study explored the correlation between pulmonary function and serum lactate dehydrogenase, and investigated thresholds for changes in pulmonary function indicators in the total population as well as in different strata of the population.

### Methods

Based on data from the National Health and Nutrition Examination Survey (NHANES) 2011–2012 (n = 3453), univariate and stratified analyses were performed to investigate factors associated with pulmonary function, and multiple regression analysis was used to further investigate the specific relationship with serum lactate dehydrogenase. Smoothed curve fitting, threshold effect and saturation effect analysis were used to explore the threshold level of serum lactate dehydrogenase at the onset of changes in pulmonary function indicators.

### Results

Adjusted smoothed curve fit plots showed a linear relationship between serum lactate dehydrogenase levels and forced vital capacity and forced expiratory volume in one second: for each 1 U/L increase in serum lactate dehydrogenase levels, forced vital capacity decreased by 1.24 mL (95% CI = -2.05, -0.42, P = 0.0030) and forced expiratory volume in one second by 1.11 mL (95% CI = -1.82, -0.39, P = 0.0025).

### Conclusions

Serum lactate dehydrogenase was negatively and linearly correlated with pulmonary function indices in the total population analyzed. Based on the total population and different population stratifications, this study determined the threshold values of serum lactate

**Funding:** This study was supported by grants from the National Natural Science Foundation of China [grant numbers 81860379, 82160410] and the Science and Technology Planning Project at the Department of Science and Technology of Jiangxi Province, China [grant number 20171BAB 205075]. The funders had no role in study design, data collection and analysis, decision to publish, or preparation of the manuscript.

**Competing interests:** The authors have declared that no competing interests exist.

**Abbreviations:** NHANES, national health and nutrition examination survey; COPD, chronic obstructive pulmonary disease; LDH, serum lactate dehydrogenase; FVC, forced vital capacity; FEV1, forced expiratory volume in the first second of expiration; PFTs, pulmonary function tests; NCHS, National Center for Health Statistics; CDC, centers for disease control and prevention; LD, lactic acid.

dehydrogenase at the onset of decline of pulmonary function in different populations. This provides a new serological monitoring indicator for patients suffering from respiratory diseases and has implications for patients with possible clinical impairment of pulmonary function. However, our cross-sectional study was not able to determine a causal relationship between these two factors, and further research is needed.

## Introduction

The leading causes of disability and death worldwide are respiratory diseases, such as chronic obstructive pulmonary disease (COPD) and asthma, respiratory viral infections, and lung cancer. Since they have a mostly chronic progressive course they are a serious social and economic burden [1, 2]. Pulmonary function tests (PFTs) are used to assess the lung status of patients over time and have become an important component of pulmonary disease assessment programs [3]. Currently, the common metrics reported in PFTs are forced expiratory volume in one-second (FEV1) and the ratio to forced vital capacity (FVC). Because pulmonary function reflects the respiratory function of an individual, it is widely used for preoperative diagnosis of respiratory disease, surgical tolerance, postoperative assessment of patient recovery, and clinical management. In clinical practice, however, PFTs are contraindicated in patients with conditions such as severe cardiovascular disease, hemoptysis, active tuberculosis, poorly controlled hypertension, recent sinus surgery or middle ear surgery or infection, recent abdominal or thoracic surgery, or inability to follow instructions [4]. In addition, although PFTs are widely available in large hospitals, they remain to be improved in primary care hospitals due to uneven development [5]. This makes it difficult for clinicians to correctly assess the pulmonary function of patients and increases the risk of misdiagnosis and missed diagnosis.

Serological indicators may be a way to indirectly assess pulmonary function: previous studies found a significant correlation between serological indicator KL-6, cysteine-rich 61 and lung function tests in patients with respiratory diseases [6, 7]. Hence, developing universal serological screening indicators may be more accurate and efficient as well as less contraindicated. Serum lactate dehydrogenase (LDH) is an important oxidoreductase enzyme of the glycolytic pathway that is widely present in human tissues and usually elevated during inflammatory processes. Previous studies have found that LDH plays an important role as an indicator of inflammation in organ damage and is also commonly used in the diagnosis of myocardial infarction [8, 9], liver disease [10, 11], and malignancy [12–14]. The relationship between lactate dehydrogenase and pulmonary function in clinical practice is currently unclear although elevated concentrations of lactate dehydrogenase have been found in the serum of COPD patients and smoking patients [15–17]. Many of the previous study populations were not representative, which may have led to an underestimation of the clinical significance of lactate dehydrogenase.

The National Health and Nutrition Examination Survey (NHANES) is a multi-phase, ongoing, representative survey conducted by the CDC to assess the health status of the U.S. population based on a large body of data [18, 19]. The rigor and reliability of NHANES data has been confirmed by numerous studies [20, 21], so data from NHANES 2011–2012 were used in this study. Our goal was to conduct an in-depth and detailed stratified study to assess the relationship between LDH and pulmonary function indicators.

## Materials and methods

### Ethics statement

This study was approved by the ethical review committee of the National Center for Health Statistics (NCHS) and the ethical review committee of the Second Clinical School of Nanchang University. Written, informed consent was obtained from the participants.

### Study population

The data for this study were obtained from NHANES III, and detailed information on the survey methodology and data collection is available on the NCHS website (http://www.cdc.gov/nchs/). Our analysis was based on data recorded from 2011 to 2012, the most recent data available for the pulmonary function indicators FVC and FEV 1. A total of 4500 individuals were included in our study. During data collation we excluded individuals with missing data on FVC, FEV 1, serum albumin levels, and LDH levels. We also excluded patients whose behavior prior to data collection could interfere with the findings, such as those collected after smoking, eating, drinking alcohol, and thirty minutes after drinking coffee. Finally, patients with data missing from their medical records such as pregnancy, history of respiratory disease, and chest surgery were also excluded. The final total was 3453 participants and the detailed process is shown in Fig 1.

### Variables

LDH was the exposure variable in this study. We divided levels into three groups: low was ≥32 to 114 U/L (n = 1123); medium was ≥114 to 133 U/L (n = 1131), and high was ≥133 to ≤491 U/L (n = 1199). These groupings were predetermined based on previous studies that found an association between LDH and respiratory function [15, 20, 21]. The outcome variables were FVC and FEV 1, which were measured based on the latest American Thoracic Society standard procedure for functional spirometry assessment. The following continuous covariates were included: age, weight (kg), standing height (cm), systolic and diastolic blood pressure (mmHg) serum glucose (mmol/L), albumin (g/L), globulin (g/L), cholesterol (mmol/L), creatinine (μmol/L), and alanine aminotransferase (ALT, U/L). The following categorical variables were included as covariates: gender, race, smoking, education level, chest or abdominal surgery, and respiratory disease. LDH was measured using LD reagent (lactic acid as substrate) DxC800 (Beckman Instruments Inc, Brea, USA), which uses an enzyme rate method to measure LD activity in biological fluids. The system monitors the rate of change of absorbance at 340 nm over a fixed time interval, which is proportional to the activity of LD in the sample. More information on LDH, FVC, FEV 1 and covariate assays is detailed at https://www.cdc.gov/nchs/nhanes/.

### Statistical analysis

SPSS v.26 (IBM Corporation, Armonk, NY, USA) and Empower Stats (https://www.empowerstats.com, X&Y Solutions, Inc., Boston, MA) were used for statistical analysis of all data. P<0.05 indicates a statistically significant difference. The relationship between LDH levels and FVC and FEV1 was analyzed according to a weighted multivariate logistic regression model. The non-linear link between lactate dehydrogenase level and FVC and FEV 1 was addressed using smooth curve fitting and a generalized additive model. We used smooth curve fitting to examine whether the independent variable was partitioned into intervals. We applied segmented regression (also known as piece-wise regression) that used a separate line segment to fit each interval. A log-likelihood ratio test comparing a one-line (non-segmented) model to a

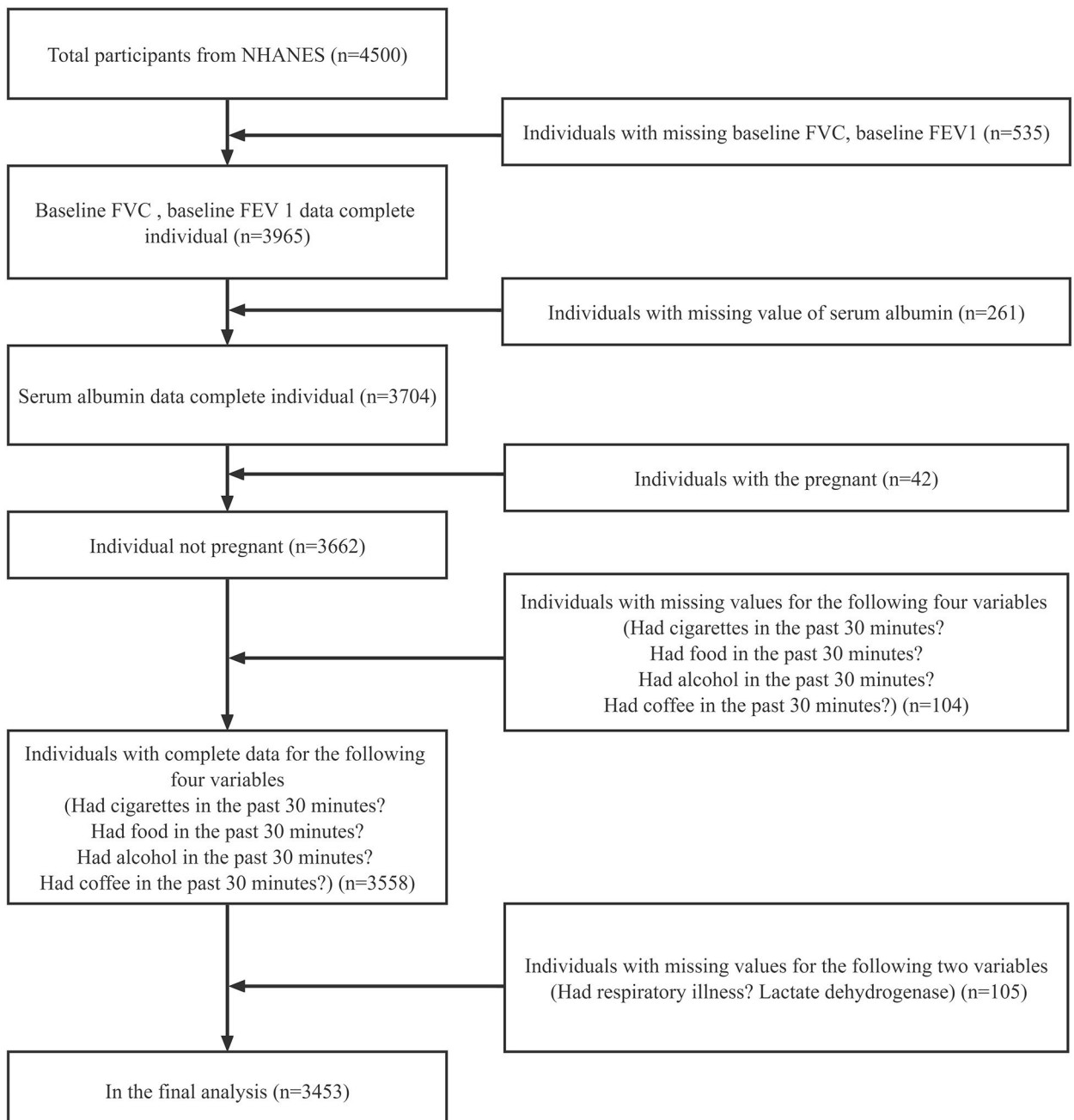

**Fig 1. Flowchart of the screening process for selecting eligible participants from NHANES 2011–2012.**

segmented regression model was used to determine whether threshold exists (when $p < 0.05$ was considered to apply to the segmented model). The inflection point that connected the segments was based on the model that gave maximum likelihood, and it was determined using a two-step recursive method. For the analysis of differences between groups, we used a weighted chi-square test for categorical data and a weighted linear regression model for continuous variables.

## Results

### Baseline characteristics of participants

People in the high lactate dehydrogenase (LDH) group were older and had higher body weight and blood pressure. The results showed that in the group with different lactate dehydrogenase levels, FVC and FEV 1 were the only variables that decreased with increasing lactate dehydrogenase levels (P<0.001). Variables with non-significant differences included gender, height, serum glucose level, and respiratory disease. Among the serological indices, globulin, cholesterol, creatinine, and ALT showed a gradual increase with lactate dehydrogenase (P<0.001). The rest of the variables with statistically significant differences are detailed in Table 1.

### Univariate and stratified analysis of the relationship between serum lactate dehydrogenase and pulmonary function

The reference group for each variable in the univariate analysis was the first group. There was a negative correlation between LDH levels and pulmonary function (Table 2, P<0.001). For the baseline FVC analysis, the beta value (CI) for LDH levels was -126.02 (-213.51, -38.53) in the middle tertile group and -332.56 (-418.80, -246.31) in the high tertile group compared to the low tertile group, both P<0.0001. For analysis of baseline FEV 1, the beta value (CI) of LDH levels was -129.34 (-201.51, -57.16) in the middle tertile group and -309.63 (-380.78, -238.48) in the high tertile group compared to the low tertile group, both P<0.0001. Age, gender, race, education level, thoracic/abdominal surgery, respiratory disease, weight, height, and systolic blood pressure were associated with FVC and FEV1 as detailed in Table 2 (P<0.05). For baseline FVC and FEV 1, differences in serum glucose and cholesterol were significant only in the higher tertile groups. Smoking was only significantly associated with FVC and not with FEV 1. Diastolic blood pressure was not significantly related to either FVC or FEV 1. Therefore, for further study, a stratified analysis was performed (S1 Table).

### Multiple regression equation analysis of the relationship between serum lactate dehydrogenase levels and pulmonary function

The results of multivariate analysis showed a negative correlation between LDH and pulmonary function (Table 3, P<0.01). In the different models, the beta values of both FVC and FEV 1 decreased progressively with increasing lactate dehydrogenase levels. In the unadjusted model, lactate dehydrogenase levels were associated with lower FVC (β = -126.02, 95% CI = -213.51, -38.53, P<0.001) and FEV 1 (β = -129.34, 95% CI = -201.51, -57.16, P<0.0001) in the intermediate subgroup compared with the low tertile group. Higher subgroup lactate dehydrogenase levels were associated with lower FVC (β = -332.56, 95% CI = -418.80, -246.31, p<0.001) and FEV 1 (β = -309.63, 95% CI = -380.78, -238.48, p<0.0001) compared to the lower tertile group. In adjusted models I and II, high lactate dehydrogenase levels were also associated with lower FVC and FEV 1 (Table 3). In fully adjusted model III, high lactate dehydrogenase levels were associated with lower FVC (β = -56.75, 95% CI = -105.43, -8.08, p<0.05) and FEV 1 (β = -53.28, 95% CI = -95.95, -10.62, p<0.05). The covariates used for adjustment in the model are detailed in Table 3.

### Smooth curve fitting, threshold effect and saturation effect analysis between serum lactate dehydrogenase levels and pulmonary function

To further clarify the relationship between LDH levels and lung function, we performed a smoothed curve fit (Fig 2) as well as threshold and saturation effect analyses (Table 4). The smoothed curve fit was adjusted to detect a nonlinear relationship, to determine the presence

**Table 1. Baseline characteristics of participants (N = 3453).**

| Lactate dehydrogenase (U/L) Tertile | Low(≥32 to 114) | Middle(≥114 to 133) | High(≥133 to ≤491) | P-value |
|---|---|---|---|---|
| Age, mean±SD (years) | 40.14 ± 14.11 | 43.00 ± 14.14 | 46.35 ± 14.02 | <0.001 |
| Weight (kg) | 78.74 ± 19.54 | 81.76 ± 20.98 | 85.36 ± 23.36 | <0.001 |
| Standing Height (cm) | 168.98 ± 9.50 | 168.55 ± 10.19 | 168.10 ± 10.13 | 0.104 |
| Systolic blood pressure (mmHg) | 117.91 ± 14.74 | 120.24 ± 15.52 | 125.51 ± 18.24 | <0.001 |
| Diastolic blood pressure (mmHg) | 70.86 ± 11.16 | 72.44 ± 11.34 | 73.91 ± 12.93 | <0.001 |
| Glucose, serum (mmol/L) | 5.53 ± 2.17 | 5.53 ± 2.04 | 5.61 ± 2.04 | 0.520 |
| Albumin (g/L) | 43.18 ± 3.25 | 43.40 ± 3.19 | 42.91 ± 3.25 | 0.001 |
| Globulin (g/L) | 28.33 ± 4.39 | 28.86 ± 4.37 | 29.41 ± 4.77 | <0.001 |
| Cholesterol (mmol/L) | 4.81 ± 0.95 | 4.99 ± 1.04 | 5.10 ± 1.12 | <0.001 |
| Creatinine (umol/L) | 74.74 ± 19.45 | 76.74 ± 23.20 | 80.73 ± 36.86 | <0.001 |
| Alanine aminotransferase ALT (U/L) | 20.81 ± 10.72 | 24.33 ± 14.26 | 30.69 ± 26.50 | <0.001 |
| Lactate dehydrogenase (U/L) | 101.28 ± 9.58 | 122.84 ± 5.37 | 153.03 ± 22.72 | <0.001 |
| Baseline FVC (mL) | 4108.70 ± 1026.93 | 3982.67 ± 1085.83 | 3776.14 ± 1064.77 | <0.001 |
| Baseline FEV 1 (mL) | 3281.56 ± 862.37 | 3152.22 ± 880.56 | 2971.93 ± 879.05 | <0.001 |
| Gender (%) | | | | 0.400 |
| Male | 50.4 | 53.2 | 52.1 | |
| Female | 49.6 | 46.8 | 47.9 | |
| Race/Hispanic origin (%) | | | | <0.001 |
| Mexican American | 10.6 | 11.2 | 10.8 | |
| Other Hispanic | 10.2 | 11.8 | 8.3 | |
| Non-Hispanic white | 39.1 | 35.1 | 30.3 | |
| Non-Hispanic black | 20.2 | 24.3 | 34.9 | |
| Other races—Including multi-racial | 19.9 | 17.6 | 15.6 | |
| Education level (%) | | | | <0.001 |
| Less than 9th grade | 5 | 7.3 | 7.6 | |
| 9-11th grade | 11.9 | 12 | 14.6 | |
| High school graduate | 17.8 | 19.2 | 22.3 | |
| Some college or AA degree | 33.9 | 32.3 | 31.4 | |
| College graduate or above | 31.3 | 29.3 | 24.2 | |
| Thoracic/abdominal surgery | | | | 0.021 |
| Yes | 16.7 | 19.5 | 21.3 | |
| No | 83.3 | 80.5 | 78.7 | |
| Respiratory disease | | | | 0.058 |
| Yes | 16.2 | 16.4 | 19.5 | |
| No | 83.8 | 83.6 | 80.5 | |
| Cigarette | | | | 0.007 |
| Yes | 3.7 | 1.9 | 1.8 | |
| No | 96.3 | 98.1 | 98.2 | |

Note: continuous variables were presented as mean±SD; categorical variables were presented as n (%). FVC: forced vital capacity; FEV1: Forced expiratory volume in one second.

or absence of a threshold effect, and the feasibility of using linear regression. The results showed a linear relationship between LDH levels and FVC and FEV 1: for each 1 U/L increase in LDH levels, FVC decreased by 1.24 mL (95% CI = -2.05, -0.42, P = 0.0030) and FEV 1 decreased by 1.11 mL (95% CI = -1.82, -0.39, P = 0.0025; Fig 2A and 2B and Table 4). The covariates used for adjustment are detailed in Table 4.

**Table 2. Crude univariate analysis for baseline FVC and baseline FEV 1.**

| Exposure | Statistics | Baseline FVC (mL) β(95%CI) P | Baseline FEV 1 (mL) β(95%CI) P |
|---|---|---|---|
| **Lactate dehydrogenase (U/L)** | 126.31 ± 25.96 | -5.68 (-7.04, -4.32) <0.0001 | -5.12 (-6.25, -4.00) <0.0001 |
| **Lactate dehydrogenase (U/L) Tertile** | | | |
| Low | 1123 (32.52%) | 0 | 0 |
| Middle | 1131 (32.75%) | -126.02 (-213.51, -38.53) 0.0048 | -129.34 (-201.51, -57.16) 0.0005 |
| High | 1199 (34.72%) | -332.56 (-418.80, -246.31) <0.0001 | -309.63 (-380.78, -238.48) <0.0001 |
| **Age (years)** | 43.23 ± 14.31 | -27.59 (-29.90, -25.27) <0.0001 | -30.87 (-32.66, -29.09) <0.0001 |
| **Age (years) Tertile** | | | |
| Low | 1135 (32.87%) | 0 | 0 |
| Middle | 1112 (32.20%) | -305.25 (-387.77, -222.72) <0.0001 | -409.34 (-473.52, -345.16) <0.0001 |
| High | 1206 (34.93%) | -909.90 (-990.78, -829.01) <0.0001 | -1017.19 (-1080.09, -954.29) <0.0001 |
| **Gender** | | | |
| Male | 1793 (51.93%) | 0 | 0 |
| Female | 1660 (48.07%) | -1337.96 (-1393.59, -1282.33) <0.0001 | -986.47 (-1035.40, -937.54) <0.0001 |
| **Race/Hispanic origin** | | | |
| Mexican American | 376 (10.89%) | 0 | 0 |
| Other Hispanic | 347 (10.05%) | -246.87 (-394.76, -98.99) 0.0011 | -182.61 (-307.22, -57.99) 0.0041 |
| Non-Hispanic white | 1199 (34.72%) | 325.31 (207.89, 442.73) <0.0001 | 147.85 (48.91, 246.80) 0.0034 |
| Non-Hispanic black | 921 (26.67%) | -493.60 (-615.18, -372.02) <0.0001 | -416.15 (-518.60, -313.70) <0.0001 |
| Other races—Including multi-racial | 610 (17.67%) | -307.81 (-438.06, -177.55) <0.0001 | -200.60 (-310.36, -90.84) 0.0003 |
| **Education level (%)** | | | |
| Less than 9th grade | 229 (6.63%) | 0 | 0 |
| 9-11th grade | 445 (12.89%) | 205.73 (36.09, 375.37) 0.0175 | 188.19 (48.15, 328.22) 0.0085 |
| High school graduate | 684 (19.81%) | 264.61 (105.36, 423.86) 0.0011 | 238.23 (106.77, 369.69) 0.0004 |
| Some college or AA degree | 1122 (32.49%) | 309.07 (157.82, 460.32) <0.0001 | 302.57 (177.72, 427.43) <0.0001 |
| College graduate or above | 973 (28.18%) | 396.50 (243.30, 549.71) <0.0001 | 372.00 (245.53, 498.47) <0.0001 |
| **Thoracic/abdominal surgery** | | | |
| Yes | 663 (19.20%) | 0 | 0 |
| No | 2790 (80.80%) | 443.22 (353.96, 532.48) <0.0001 | 426.33 (352.89, 499.77) <0.0001 |
| **Respiratory disease** | | | |
| Yes | 601 (17.41%) | 0 | 0 |
| No | 2852 (82.59%) | 151.43 (57.58, 245.29) 0.0016 | 151.71 (74.16, 229.25) 0.0001 |
| **Cigarette** | | | |
| Yes | 85 (2.46%) | 0 | 0 |
| No | 3368 (97.54%) | -316.59 (-546.33, -86.84) 0.0069 | -177.82 (-367.87, 12.23) 0.0668 |
| **Weight (kg)** | 82.02 ± 21.56 | 10.47 (8.85, 12.09) <0.0001 | 7.22 (5.87, 8.57) <0.0001 |
| **Weight (kg) Tertile** | | | |
| Low | 1143 (33.27%) | 0 | 0 |
| Middle | 1144 (33.29%) | 465.97 (380.93, 551.01) <0.0001 | 309.38 (238.40, 380.36) <0.0001 |
| High | 1149 (33.44%) | 595.51 (510.57, 680.46) <0.0001 | 410.99 (340.09, 481.89) <0.0001 |
| **Standing Height (cm)** | 168.53 ± 9.95 | 78.59 (76.15, 81.03) <0.0001 | 58.60 (56.37, 60.82) <0.0001 |
| **Standing Height (cm) Tertile** | | | |
| Low | 1136 (33.05%) | 0 | 0 |
| Middle | 1150 (33.46%) | 784.13 (719.90, 848.36) <0.0001 | 572.14 (515.04, 629.23) <0.0001 |
| High | 1151 (33.49%) | 1775.87 (1711.65, 1840.08) <0.0001 | 1327.41 (1270.33, 1384.49) <0.0001 |
| **Systolic blood pressure (mmHg)** | 121.31 ± 16.58 | -7.56 (-9.73, -5.38) <0.0001 | -8.76 (-10.55, -6.97) <0.0001 |
| **Systolic blood pressure (mmHg) Tertile** | | | |
| Low | 1050 (31.70%) | 0 | 0 |

*(Continued)*

**Table 2.** (Continued)

| Exposure | Statistics | Baseline FVC (mL) β(95%CI) P | Baseline FEV 1 (mL) β(95%CI) P |
|---|---|---|---|
| Middle | 1102 (33.27%) | 213.46 (124.15, 302.77) <0.0001 | 124.48 (50.91, 198.04) 0.0009 |
| High | 1160 (35.02%) | -137.59 (-225.80, -49.38) 0.0023 | -229.84 (-302.50, -157.18) <0.0001 |
| **Diastolic blood pressure (mmHg)** | 72.44 ± 11.92 | 3.59 (0.54, 6.63) 0.0210 | 0.44 (-2.08, 2.97) 0.7298 |
| **Diastolic blood pressure (mmHg) Tertile** | | | |
| Low | 988 (29.83%) | 0 | 0 |
| Middle | 1210 (36.53%) | 66.21 (-23.38, 155.80) 0.1476 | 4.71 (-69.49, 78.91) 0.9010 |
| High | 1114 (33.64%) | 84.63 (-6.69, 175.94) 0.0694 | -17.95 (-93.57, 57.68) 0.6419 |
| **Glucose, serum (mmol/L)** | 5.56 ± 2.09 | -60.91 (-77.88, -43.94) <0.0001 | -58.41 (-72.40, -44.41) <0.0001 |
| **Glucose, serum (mmol/L) Tertile** | | | |
| Low | 1113 (32.23%) | 0 | 0 |
| Middle | 1119 (32.41%) | -35.03 (-123.10, 53.05) 0.4358 | -46.19 (-118.57, 26.20) 0.2112 |
| High | 1221 (35.36%) | -273.30 (-359.52, -187.08) <0.0001 | -311.36 (-382.23, -240.50) <0.0001 |
| **Albumin (g/L)** | 43.16 ± 3.24 | 114.79 (104.47, 125.10) <0.0001 | 102.05 (93.62, 110.48) <0.0001 |
| **Albumin (g/L) Tertile** | | | |
| Low | 1027 (29.74%) | 0 | 0 |
| Middle | 1257 (36.40%) | 389.15 (305.67, 472.63) <0.0001 | 323.37 (255.00, 391.74) <0.0001 |
| High | 1169 (33.85%) | 854.16 (769.28, 939.03) <0.0001 | 758.62 (689.11, 828.14) <0.0001 |
| **Globulin (g/L)** | 28.88 ± 4.54 | -67.87 (-75.40, -60.34) <0.0001 | -47.61 (-53.91, -41.31) <0.0001 |
| **Globulin (g/L) Tertile** | | | |
| Low | 1030 (29.88%) | 0 | 0 |
| Middle | 1002 (29.07%) | -314.50 (-403.97, -225.03) <0.0001 | -238.57 (-313.31, -163.83) <0.0001 |
| High | 1415 (41.05%) | -691.55 (-774.13, -608.96) <0.0001 | -491.29 (-560.28, -422.30) <0.0001 |
| **Cholesterol (mmol/L)** | 4.97 ± 1.05 | -107.05 (-140.91, -73.20) <0.0001 | -111.59 (-139.49, -83.69) <0.0001 |
| **Cholesterol (mmol/L) Tertile** | | | |
| Low | 1130 (32.73%) | 0 | 0 |
| Middle | 1163 (33.68%) | -43.84 (-130.96, 43.29) 0.3241 | -56.88 (-128.69, 14.93) 0.1207 |
| High | 1160 (33.59%) | -224.22 (-311.40, -137.04) <0.0001 | -246.06 (-317.92, -174.20) <0.0001 |
| **Creatinine (umol/L)** | 77.47 ± 27.87 | 6.47 (5.21, 7.73) <0.0001 | 4.38 (3.33, 5.42) <0.0001 |
| **Creatinine (umol/L) Tertile** | | | |
| Low | 1121 (32.46%) | 0 | 0 |
| Middle | 1161 (33.62%) | 602.70 (520.03, 685.37) <0.0001 | 434.11 (364.69, 503.52) <0.0001 |
| High | 1171 (33.91%) | 853.80 (771.30, 936.30) <0.0001 | 609.69 (540.42, 678.96) <0.0001 |
| **Alanine aminotransferase ALT (U/L)** | 25.39 ± 19.09 | 7.24 (5.39, 9.09) <0.0001 | 5.01 (3.48, 6.55) <0.0001 |
| **Alanine aminotransferase ALT (U/L) Tertile** | | | |
| Low | 1140 (33.02%) | 0 | 0 |
| Middle | 1112 (32.21%) | 273.95 (187.55, 360.36) <0.0001 | 170.55 (98.69, 242.40) <0.0001 |
| High | 1200 (34.76%) | 528.56 (443.77, 613.34) <0.0001 | 373.70 (303.19, 444.20) <0.0001 |

Note: continuous variables were presented as mean±SD; categorical variables were presented as n (%). The first group was used as the reference (β = 0) for each univariate analysis group; (a) including multi-Racial; (b) includes 12th grade with no diploma; (c) GED or equivalent. Weighted by: full sample mobile examination center exam weight. Abbreviations: FVC: forced vital capacity; FEV1, forced expiratory volume in one second.

## Smoothed curve fitting for each factor stratification, threshold effect and saturation effect analysis

Smooth-fit curves were plotted for the relationship between the different strata of the six covariates and LDH levels (Figs 3 and 4). For more detailed analysis, threshold effect and saturation effect analyses were performed to clarify the changes in FEV and FEV 1 with increasing

**Table 3. Relationship between serum and serum lactate dehydrogenase and pulmonary function (multiple regression equation analysis).**

| Outcome | Rough model β (95%CI) P-value | Model I β (95%CI) P-value | Model II β (95%CI) P-value | Model III β (95%CI) P-value |
|---|---|---|---|---|
| **Y = Baseline FVC (mL)** | | | | |
| **Lactate dehydrogenase (U/L)** | -5.68 (-7.04, -4.32) <0.0001 | -3.65 (-4.61, -2.69) <0.0001 | -2.52 (-3.41, -1.64) <0.0001 | -1.24 (-2.05, -0.42) 0.0030 |
| **Lactate dehydrogenase (U/L) Tertile** | | | | |
| Low | 0 | 0 | 0 | 0 |
| Middle | -126.02 (-213.51, -38.53) 0.0048 | -90.41 (-151.52, -29.29) 0.0038 | -67.63 (-123.22, -12.03) 0.0172 | -35.67 (-82.16, 10.82) 0.1328 |
| High | -332.56 (-418.80, -246.31) <0.0001 | -196.47 (-257.48, -135.47) <0.0001 | -131.46 (-187.38, -75.53) <0.0001 | -56.75 (-105.43, -8.08) 0.0224 |
| **Y = Baseline FEV 1 (mL)** | | | | |
| **Lactate dehydrogenase (U/L)** | -5.12 (-6.25, -4.00) <0.0001 | -2.66 (-3.42, -1.89) <0.0001 | -1.87 (-2.61, -1.14) <0.0001 | -1.11 (-1.82, -0.39) 0.0025 |
| **Lactate dehydrogenase (U/L) Tertile** | | | | |
| Low | 0 | 0 | 0 | 0 |
| Middle | -129.34 (-201.51, -57.16) 0.0005 | -72.77 (-121.66, -23.88) 0.0036 | -58.56 (-104.81, -12.31) 0.0131 | -43.40 (-84.15, -2.65) 0.0369 |
| High | -309.63 (-380.78, -238.48) <0.0001 | -143.64 (-192.45, -94.83) <0.0001 | -98.52 (-145.05, -51.99) <0.0001 | -53.28 (-95.95, -10.62) 0.0144 |

Abbreviations: FVC: forced vital capacity; FEV1: forced expiratory volume in one second. Weighted by: full sample mobile examination center exam weight. Outcome variable: baseline FVC; baseline FEV 1. Exposure variable: lactate dehydrogenase (U/L). Rough model: variables unadjusted. Model I adjusted by gender, age; Model II adjusted by: gender, age, race; Model III adjusted by: age; gender; race/Hispanic origin; education level; thoracic/abdominal surgery (yes, no); respiratory disease (yes, no); cigarette (yes, no); weight; standing height; systolic blood pressure; diastolic blood pressure; glucose, serum; albumin; globulin; cholesterol; creatinine; alanine aminotransferase.

LDH in the different strata of each covariate. A log-likelihood ratio of <0.05 in the table indicated that a segmented model was applicable. The k value is the turning point value, i.e., the level at which the relationship between LDH and lung function will probably change. Model II is not applicable when the relationship between LDH and the outcome variable shows a linear effect.

In men, LDH was linearly negatively correlated with both FVC (β = -1.57, 95% CI = -2.85 to -0.28, P = 0.017) and FEV1 (β = -1.75, 95% CI = -2.88 to -0.61, P = 0.026). In women, LDH was linearly and negatively correlated with FVC (β = -1.14, 95% CI = -2.09 to -0.18, P = 0.0202; Figs 3A and 4A and S2 Table). In people <60 years, LDH was linearly and negatively correlated with FVC (β = -2.07, 95% CI = -3.04 to -1.10, P<0.001). In addition, the relationship between LDH and FEV1 showed a segmental effect, with a negative correlation at levels >122 U/L (β = -4.14, 95% CI = -6.18 to -2.11, P<0.001). In those aged >60 years, the relationship between LDH and FVC showed a segmental effect with a negative correlation at LDH levels >163 U/L (β = -7.14, 95% CI = -13.14 to -0.81, P = 0.0275; Figs 3B and 4B and S3 Table).

There was a linear negative association with FVC among Mexican Americans (β = -2.98, 95% CI = -5.44 to -0.53, P = 0.0178) and non-Hispanic Black participants (β = -1.52, 95% CI = -2.82 to -0.23, P = 0.0216). LDH was negatively and linearly associated with FEV1 in Mexican Americans (β = -2.52, 95% CI = -4.42 to -0.62, P = 0.0098) and non-Hispanic Black participants (β = -1.21, 95% CI = -2.41 to -0.01, P = 0.0477). In non-Hispanic White participants (β = -1.21, 95% CI = -2.41 to -0.01, P = 0.0477), LDH had a segmental effect with FEV1, with a negative correlation when levels were <132 U/L (β = -3.44, 95% CI = -5.86 to -1.02, P = 0.0054; Figs 3C and 4C and S4 Table).

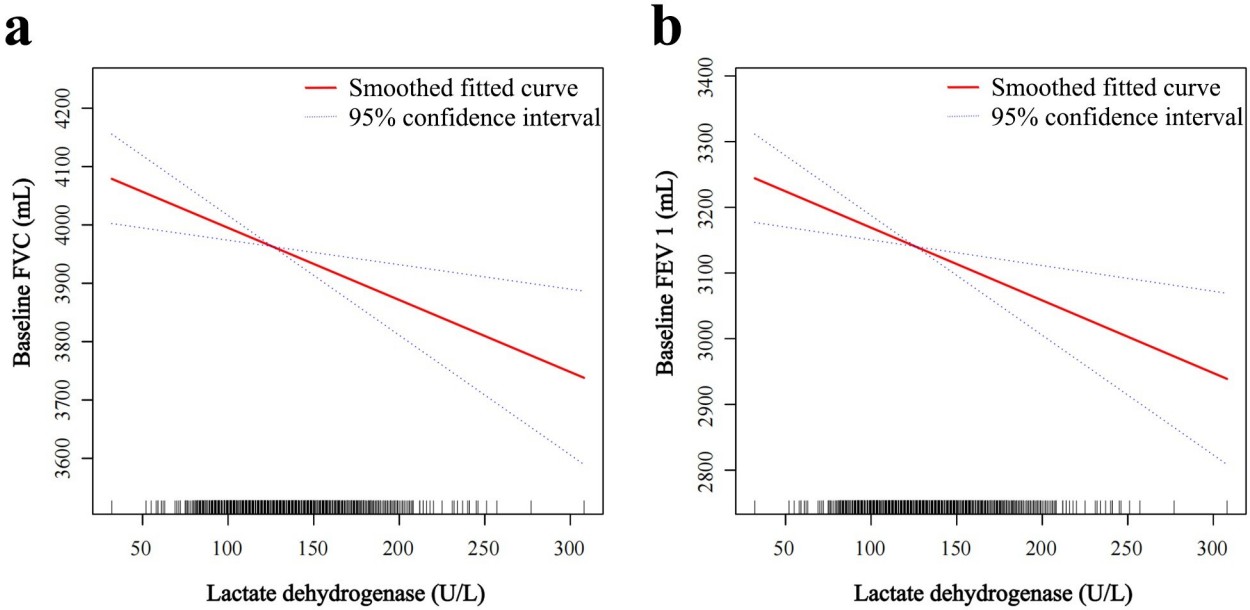

**Fig 2. Association between serum lactate dehydrogenase and pulmonary function indicators FVC and FEV1.** The red line represents the smoothed curve fit between the variables. (a) Solid line plots of curve fits for baseline lactate dehydrogenase and FVC for the main variables. (b) Solid line plots of curve fits for the primary variable between baseline lactate dehydrogenase and FEV 1. The blue line represents the 95% confidence interval of the fit. Full sample mobile examination center exam weight. Adjusted for age (smooth), sex, education, race, surgery (yes, no), respiratory disease (yes, no), cigarettes (yes, no), weight (smooth), standing height (smooth), diastolic blood pressure (smooth), systolic blood pressure (smooth), glucose, serum (smooth), cholesterol (smooth), creatinine (smooth), alanine aminotransferase (smooth), albumin (smooth), globulin (smooth).

**Table 4. Analysis of threshold effect and saturation effect.**

| Outcome | Baseline FVC (mL) β (95%CI) P-value | Baseline FEV 1 (mL) β (95%CI) P-value |
|---|---|---|
| **Model I** | | |
| A straight-line effect | -1.24 (-2.05, -0.42) 0.0030 | -1.11 (-1.82, -0.39) 0.0025 |
| **Model II** | | |
| Fold points (K) | 93 | 96 |
| < K-segment effect 1 | 4.53 (-2.44, 11.50) 0.2027 | 0.86 (-4.36, 6.07) 0.7474 |
| >K-segment Effect 2 | -1.46 (-2.32, -0.60) 0.0009 | -1.21 (-1.98, -0.44) 0.0020 |
| Effect size difference of 2 versus 1 | -5.99 (-13.18, 1.20) 0.1026 | -2.07 (-7.50, 3.37) 0.4564 |
| Equation predicted values at break points | 4172.99 (4109.00, 4236.98) | 3310.64 (3259.56, 3361.73) |
| Log likelihood ratio tests | 0.101 | 0.455 |

Abbreviations: FVC: forced vital capacity; FEV1, forced expiratory volume in one second. Weighted by: full sample mobile examination center exam weight. Outcome variable: baseline FVC, baseline FEV 1. Exposure variable: lactate dehydrogenase. Adjusted for age, gender, race/Hispanic origin, education level, thoracic/abdominal surgery, respiratory disease, cigarette, weight, standing height, systolic blood pressure, diastolic blood pressure, glucose, serum, albumin, globulin, cholesterol, creatinine, alanine aminotransferase. When P<0.05 in Model I, the model showed a straight-line effect. When P>0.05 in Model I, the model showed a segmented effect in Model II, with the K value being the lactate dehydrogenase level at the fold point; β represents the slope of the curve, β for segments with P<0.05 was statistically significant. The K value is the inflection point, which is the level of lactate dehydrogenase content at which the relationship between lactate dehydrogenase and lung function changes.

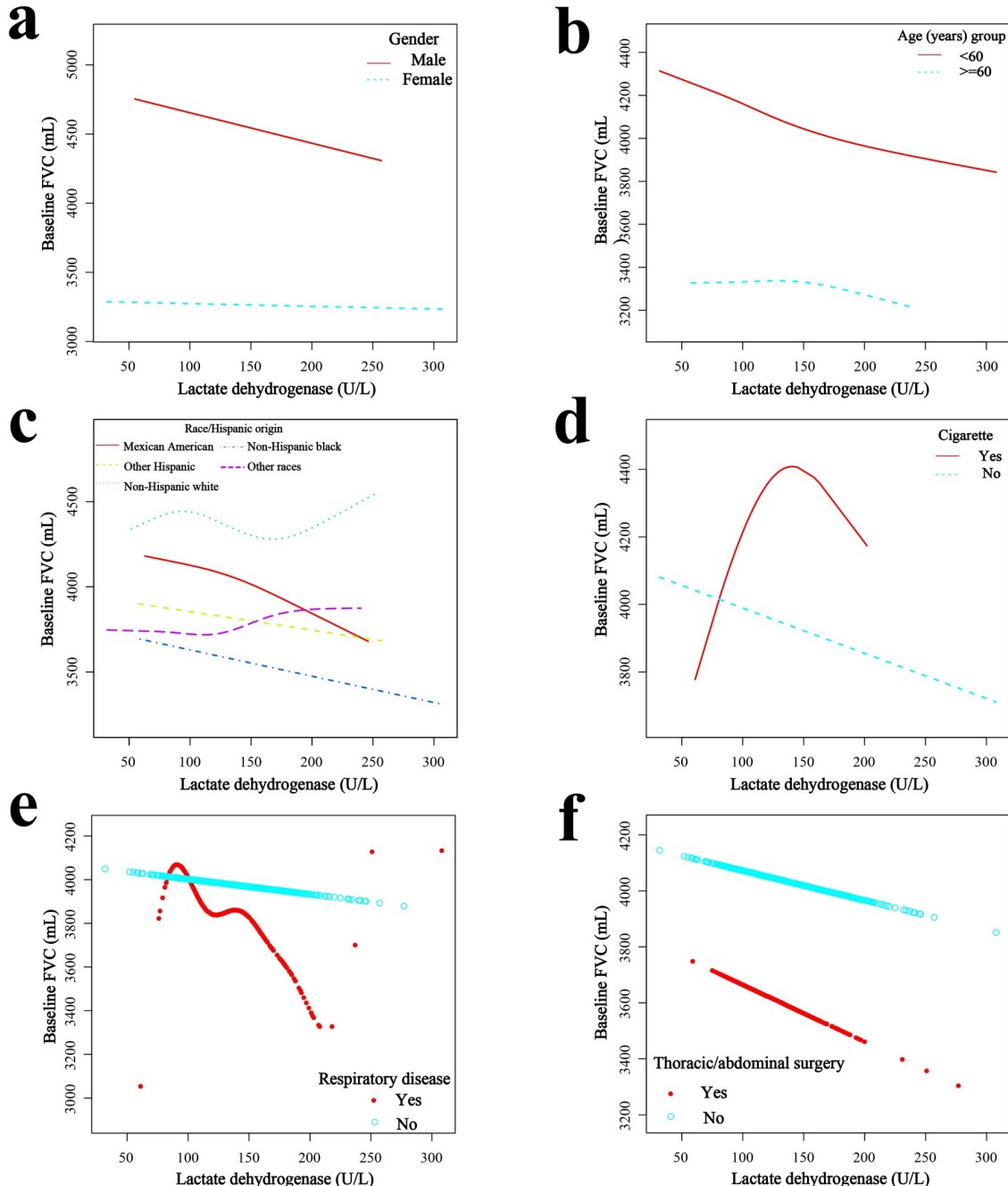

**Fig 3. Relationship between serum lactate dehydrogenase and FVC.** (a) Stratified by sex. (b) Stratified by age. (c) Stratified by race. (d) Stratified by smoking status. (e) Stratified by respiratory disease. (f) Stratified by surgery.

Among non-smokers, LDH was linearly and negatively correlated with both FVC ($\beta$ = -1.33, 95% CI = -2.15 to -0.51, P = 0.0015) and FEV1 ($\beta$ = -1.11, 95% CI = -1.83 to -0.39, P = 0.0026; Figs 3D and 4D and S5 Table).

In those with previous respiratory disease, LDH was linearly negatively associated with both FVC ($\beta$ = -2.94, 95% CI = -4.86 to -1.03, P = 0.0028) and FEV 1 ($\beta$ = -2.32, 95% CI = -4.11 to -0.52, P = 0.0116). In those without respiratory disease, serum albumin was linearly

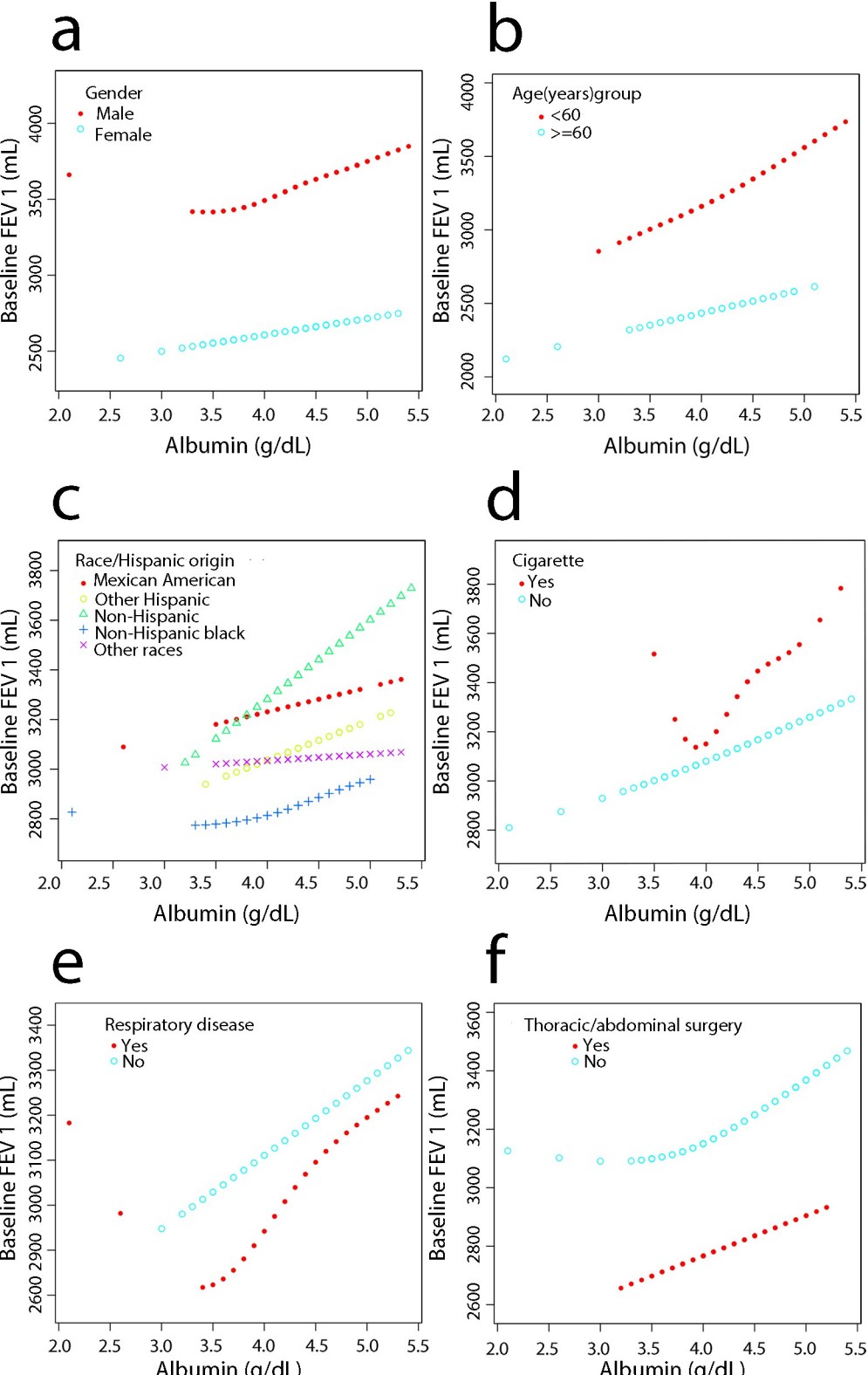

**Fig 4. Relationship between serum albumin and FEV 1.** (a) Stratified by sex. (b) Stratified by age. (c) Stratified by race. (d) Stratified by smoking status. (e) Stratified by respiratory disease. (f) Stratified by surgery.

negatively correlated only with FEV 1 (β = -0.79, 95% CI = -1.57 to -0.01, P = 0.0471; Figs 3E and 4E and S6 Table).

In those without chest or abdominal surgery, LDH was linearly negatively correlated with both FVC (β = -1.03, 95% CI = -1.94 to -0.12, P = 0.0268) and FEV1 (β = -1.09, 95% CI = -1.89 to -0.29, P = 0.0075). In contrast, in those who had previous chest or abdominal surgery, LDH levels were linearly associated with FVC (β = -2.31, 95% CI = -4.14 to -0.49, P = 0.0134). There was a segmental effect with FEV1, which was negatively associated until levels were <113 U/L (β = -6.67, 95% CI = -12.03 to -1.31, P = 0.0150; Figs 3F and 4F and S7 Table).

## Discussion

Respiratory diseases remain a significant cause of morbidity and mortality worldwide. Ongoing research continues to improve diagnostic tools and treatment options [22–25], and current clinical investigations can be divided into laboratory and specific tests. Numerous previous studies have demonstrated the value of PFTs for clinical applications [26–28], such as in patients with COPD [29] and asthma [30]. However PFTs are not suitable for all patients [4]. For example, while they are not contraindicated in patients with tracheotomy or Morquio syndrome, performing PFTs is difficult and the results are not reliable [31, 32]. Furthermore, in the current phase of the COVID-19 epidemic, PFTs may be a potential route of transmission because of the aerosols generated during the procedure and the concentration of patients with pulmonary disease in the laboratory [33].

Serologic indicators are more universal than PFTs, have fewer contraindications, and can accurately and efficiently reflect relevant information about the sample. Previous studies have confirmed that more and more serologic markers are being used to diagnose and monitor diseases such as cancer, COVID-19, and cardiac diseases [34–38]. In addition, a large number of studies describe indirect associations between serologic indicators and pulmonary function [7, 15, 39], but the status of these indicators in the diagnosis and treatment of respiratory diseases needs to be further improved. The NHANES database has been used in many studies, and is a well-collected and representative population [40–43]. Hence, we obtained a large amount of valuable serological index data from this database for analysis and determined the potential value of LDH.

LDH, an important inflammatory marker, is underestimated in terms of its clinical significance [44]. Previous studies have suggested that LDH levels are associated with lung disease [45]. In recent years, it has not only been shown to be a prognostic marker for diseases such as non-small cell lung cancer [46], idiopathic pulmonary fibrosis [47], and metastatic breast cancer [48], but is also a common indicator in diagnosis [8–14]. In fact, LDH levels have important implications in pulmonary disease activity and response to therapy. Mura et al. plasma LDH was found to be induced by hypoxia and LDH levels were found to be increased in 22 patients diagnosed with IPF, but the relationship between LDH and IPF severity was unclear [49]. Spruit et al. showed that increased muscle LDH activity was found in older men with COPD and that resting serum LDH activity was increased in COPD patients compared to healthy smoking and non-smoking peers [50]. However, the relationship between LDH levels and pulmonary function was unclear. Previous studies have suggested that an inflammatory response due to impaired pulmonary function may be responsible for elevated levels [51]. LDH is present in cells, and when lung injury or inflammation decreases pulmonary function, LDH released from the cells increases serum levels. A previous study found lower indicators of pulmonary function and elevated serum LDH in patients with COPD relative to healthy patients [15]. Our results also showed a negative correlation between serum LDH and pulmonary function. Although serum LDH levels are not exactly equivalent to tissue LDH levels,

tissue-level LDH expression may correlate with serum LDH levels [52]. Previous studies found that high expression of LDH in cancer mediates tumor immune escape leading to tumorigenesis or progression by suppressing the killing effect of immunity and promoting the suppressive effect of immunity. Thus, serum LDH may also indicate decreased pulmonary function due to progression of some respiratory cancers [53–56]. These studies provide guidance for future monitoring of serum LDH levels in response to changes in pulmonary function and predicting respiratory failure in specific populations.

Our study still has some limitations. The data from NHANES 2011–2012 are the most recent and representative data available that contain indicators of pulmonary function. In addition, although our sample size has improved compared to previous studies, data from a larger number of participants would have made the findings more convincing. Our cross-sectional study cannot mechanistically determine the causal relationship between these two factors, and further research is needed [57]. Although we controlled for confounding factors by statistical methods, we still may not be able to exclude the interference of other confounding factors. Hence, if more data are obtained or supported by more prospective and mechanistic studies, we believe that the relationship between LDH and pulmonary function will be more deeply interpreted in the future.

## Conclusions

The relationship between the serum marker lactate dehydrogenase and pulmonary function was explored in a large number of cases and in more detailed population stratification than previous studies. LDH levels were negatively correlated with pulmonary function. This study provides a new way to monitor changes in pulmonary function in patients for whom PFTs are clinically contraindicated. This provides a theoretical basis for lactate dehydrogenase as an indicator of pulmonary function. Identifying a threshold for LDH when PFTs begin to decline provides guidance for the diagnosis of respiratory disease.

## Supporting information

**S1 Table. Stratification analysis between serum lactate dehydrogenase and baseline FVC, serum lactate dehydrogenase and baseline FEV 1.** (a) including Multi-Racial; (b) includes 12th grade with no diploma; (c) GED or equivalent. Weighted by: full sample mobile examination center exam weight.
(DOCX)

**S2 Table. Analysis of threshold effect and saturation effect (Stratification by gender).**
(DOCX)

**S3 Table. Analysis of threshold effect and saturation effect (Stratification by age).**
(DOCX)

**S4 Table. Analysis of threshold effect and saturation effect (Stratification by race/Hispanic origin).**
(DOCX)

**S5 Table. Analysis of threshold effect and saturation effect (Stratification by cigarette).**
(DOCX)

**S6 Table. Analysis of threshold effect and saturation effect (Stratification by respiratory disease).**
(DOCX)

**S7 Table. Analysis of threshold effect and saturation effect (Stratification by thoracic/abdominal surgery).**
(DOCX)

**S1 Raw data.**
(XLSX)

## Acknowledgments

We hereby thank the participants for their time and energy in the data collection phase of NHANES.

## Author Contributions

**Conceptualization:** Sheng Hu, Jiayue Ye, Qiang Guo, Yiping Wei.

**Data curation:** Sheng Hu, Jiayue Ye, Qiang Guo.

**Formal analysis:** Sheng Hu, Jiayue Ye, Qiang Guo, Lang Su.

**Funding acquisition:** Yiping Wei.

**Investigation:** Sheng Hu, Jiayue Ye, Qiang Guo, Deyuan Zhang, Yang Zhang, Silin Wang, Lang Su.

**Methodology:** Sheng Hu, Jiayue Ye, Qiang Guo, Yiping Wei.

**Project administration:** Sheng Hu, Yiping Wei.

**Resources:** Sheng Hu, Jiayue Ye, Qiang Guo.

**Software:** Sheng Hu, Jiayue Ye, Qiang Guo.

**Supervision:** Yiping Wei.

**Validation:** Sheng Zou, Wenxiong Zhang, Deyuan Zhang, Yang Zhang, Silin Wang, Lang Su.

**Visualization:** Sheng Hu, Jiayue Ye, Qiang Guo.

**Writing – original draft:** Sheng Hu, Jiayue Ye, Qiang Guo.

**Writing – review & editing:** Jiayue Ye.

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
