## [Decision Letter · Decision Letter 0]

6 Dec 2022

PONE-D-22-29185Serum lactate dehydrogenase is associated with impaired lung function: NHANES 2011-2012PLOS ONE

Dear Dr. Wei,

Thank you for submitting your manuscript to PLOS ONE. After careful consideration, we feel that it has merit but does not fully meet PLOS ONE’s publication criteria as it currently stands. Therefore, we invite you to submit a revised version of the manuscript that addresses the points raised during the review process.

ACADEMIC EDITOR: Authors should revise their manuscript according to the suggestions of the reviewers. There is some novelty but the paper must be improved, accordingly.

We look forward to receiving your revised manuscript.

Kind regards,

Gulali Aktas

Academic Editor

PLOS ONE

Journal Requirements:

"This study was supported by grants from the National Natural Science Foundation of China [grant numbers 81860379, 82160410] and the Science and Technology Planning Project at the Department of Science and Technology of Jiangxi Province, China [grant number 20171BAB 205075]."

"This study was supported by grants from the National Natural Science Foundation of China [grant numbers 81860379, 82160410] and the Science and Technology Planning Project at the Department of Science and Technology of Jiangxi Province, China [grant number 20171BAB 205075]."

"This study was supported by grants from the National Natural Science Foundation of China [grant numbers 81860379, 82160410] and the Science and Technology Planning Project at the Department of Science and Technology of Jiangxi Province, China [grant number 20171BAB 205075]."

Additional Editor Comments:

Authors should revise their manuscript according to the suggestions of the reviewers. There is some novelty but the paper must be improved, accordingly.

Reviewers' comments:

Reviewer's Responses to Questions

**Comments to the Author**

1. Is the manuscript technically sound, and do the data support the conclusions?

Reviewer #1: Yes

Reviewer #2: Yes

2. Has the statistical analysis been performed appropriately and rigorously? 

Reviewer #1: Yes

Reviewer #2: Yes

3. Have the authors made all data underlying the findings in their manuscript fully available?

Reviewer #1: Yes

Reviewer #2: Yes

4. Is the manuscript presented in an intelligible fashion and written in standard English?

Reviewer #1: Yes

Reviewer #2: Yes

5. Review Comments to the Author

Reviewer #1: Increased muscle LDH activity has been found in elderly male patients with chronic obstructive pulmonary disease (COPD) who are susceptible to contractile fatigue of the quadriceps femoris muscle following exercise. (Spruit, M. A., Pennings, H. J., Möller, G. M., Janssen, P. P., & Wouters, E. F. M. (2008). Serum LDH and exercise capacity in COPD. Thorax, 63(5), 472-472 )

Plasma lactate dehydrogenase (LDH) can be induced by hypoxia and is found to be increased in multiple patients diagnosed with IPF, but its relation to IPF severity is less known.( Åttingsberg, E., Hoyer, N., Wilcke, T., Prior, T. S., Bendstrup, E., & Shaker, S. (2019). Lactate dehydrogenase as a biomarker of advanced disease in idiopathic pulmonary fibrosis)…..

Since the correlation between LDH level and PFT is examined, LDH studies in lung diseases should be given more attention.The role of LDH level in disease activity and response to treatment in lung diseases can also be mentioned. Studies can also be mentioned about the increase in LDH level with smoking.

Reviewer #2: The authors suggest the lactate dehydrogenase as a new serological monitoring indicator for patients

suffering from respiratory diseases and has implications for patients with possible

clinical impairment of pulmonary function. The manuscript Eur Respir J

. 1996 Aug;9(8):1736-42. doi: 10.1183/09031936.96.09081736 evidences the LDH involvement with lung diseases. However, the present study addressed the LDH role on worse of pulmonary ventilation in lung diseases. It is a new information.

6. PLOS authors have the option to publish the peer review history of their article (what does this mean?). If published, this will include your full peer review and any attached files.

Reviewer #1: No

Reviewer #2: No

---

## [Author Response · Author response to Decision Letter 0]

7 Dec 2022

Dear Gulali Aktas and Reviewers:

Thank you for your letter and for the comments concerning our manuscript entitled “Serum lactate dehydrogenase is associated with impaired lung function: NHANES 2011-2012”. We are very sorry for submitting the revised manuscript so late. The reviewers’ comments are all valuable and helpful for revising and improving our paper, as well providing important guiding significance to our research. We have studied the comments carefully and have made corrections that we hope will meet with approval. The main corrections of this article and the point-by-point responses to the editor and reviewers’ comments are detailed below. 

Editor:

1.- Please ensure that your manuscript meets PLOS ONE's style requirements, including those for file naming.

Response: Thank you for your professional advice. We have revised the manuscript according to PLOS ONE style requirements, including the requirements for file naming. See the revised manuscript for details.

2.- Please state what role the funders took in the study. If the funders had no role, please state: "The funders had no role in study design, data collection and analysis, decision to publish, or preparation of the manuscript." If this statement is not correct you must amend it as needed. Please include this amended Role of Funder statement in your cover letter.

Response: Thank you for your careful reading of our manuscript. We have clarified in our newly uploaded cover letter that funders have no role in study design, data collection and analysis, decision to publish, or preparation of the manuscript.

3.- Please remove any funding-related text from the manuscript and let us know how you would like to update your Funding Statement. Please include your amended statements within your cover letter.

Response: Thank you for your professional comments concerning our manuscript. We have removed any grant-related text from the manuscript and have included a corrected grant statement in the cover letter.

4.- Please review your reference list to ensure that it is complete and correct.

Response: Thank you for carefully reading our manuscript. Based on your comments, we have corrected the cited sources in reference 6. Based on your professional opinion, we found that some of the withdrawn low-quality literature was incorrectly cited. Therefore, we have removed the original references 29, 30 and 45 from the manuscript. Based on your and the reviewers' valuable comments, we have added citations 16, 17, 45, 49 and 50. See the reference section of the manuscript for details.

Reviewer #1:

Increased muscle LDH activity has been found in elderly male patients with chronic obstructive pulmonary disease (COPD) who are susceptible to contractile fatigue of the quadriceps femoris muscle following exercise. (Spruit, M. A., Pennings, H. J., Möller, G. M., Janssen, P. P., & Wouters, E. F. M. (2008). Serum LDH and exercise capacity in COPD. Thorax, 63(5), 472-472 )

Plasma lactate dehydrogenase (LDH) can be induced by hypoxia and is found to be increased in multiple patients diagnosed with IPF, but its relation to IPF severity is less known.( Åttingsberg, E., Hoyer, N., Wilcke, T., Prior, T. S., Bendstrup, E., & Shaker, S. (2019). Lactate dehydrogenase as a biomarker of advanced disease in idiopathic pulmonary fibrosis)

Since the correlation between LDH level and PFT is examined, LDH studies in lung diseases should be given more attention.The role of LDH level in disease activity and response to treatment in lung diseases can also be mentioned. Studies can also be mentioned about the increase in LDH level with smoking.

Response: Thank you for your professional comments. All these references are very valuable to our article. We have modified the article as follows based on your valuable comments. 

1) Added lines 319-320 “Previous studies have suggested that LDH levels are associated with lung disease. In recent years,”

2) Added lines 323-329 “In fact, LDH levels have important implications in pulmonary disease activity and response to therapy……Spruit et al. showed that increased muscle LDH activity was found in older men with COPD and that resting serum LDH activity was increased in COPD patients compared to healthy smoking and non-smoking peers”.

3) Added lines 75-78 “The relationship between……in the serum of COPD patients and smoking patients”.

4) Added citations for references 16, 17, 49, 50.

Thank you for considering our manuscript and putting forward such professional and constructive opinions to help us improve the quality of the manuscript.

Reviewer #2: The authors suggest the lactate dehydrogenase as a new serological monitoring indicator for patients

suffering from respiratory diseases and has implications for patients with possible clinical impairment of pulmonary function. The manuscript Eur Respir J. 1996 Aug;9(8):1736-42. doi: 10.1183/09031936.96.09081736 evidences the LDH involvement with lung diseases. However, the present study addressed the LDH role on worse of pulmonary ventilation in lung diseases. It is a new information.

Response: Thank you very much for your professional opinion and for the superb summary of the main elements of our study. Based on your valuable comments, we have cited reference 45 and the added lines 319-320 “Previous studies have suggested that LDH levels are associated with lung disease. In recent years”. Thank you for considering our study to be somewhat new and for providing such a valuable reference.

Thank you again for considering our manuscript and putting forward such professional and constructive opinions to help us improve the quality of the manuscript.

Thank you again for your careful reading of our manuscript and professional comments. We tried our best to improve the manuscript and have made some important changes to the manuscript. 

However, the changes have not influenced the overall framework of the paper. We appreciate the efforts made by the Editors and Reviewers, and hope that the corrections will meet with their approval. We hope to receive a decision soon.

Thank you and best wishes

Yours sincerely

Corresponding author:

Name: Yiping Wei

E-mail: ndefy08025@ncu.edu.cn

---

## [Decision Letter · Decision Letter 1]

18 Jan 2023

Serum lactate dehydrogenase is associated with impaired lung function: NHANES 2011-2012

PONE-D-22-29185R1

Dear Dr. Wei,

We’re pleased to inform you that your manuscript has been judged scientifically suitable for publication and will be formally accepted for publication once it meets all outstanding technical requirements.

Kind regards,

Gulali Aktas

Academic Editor

PLOS ONE

Additional Editor Comments (optional):

The paper is revised in accordance with the suggestions of the reviewers. It is acceptable for publication in its current form.

Reviewers' comments:

Reviewer's Responses to Questions

**Comments to the Author**

1. If the authors have adequately addressed your comments raised in a previous round of review and you feel that this manuscript is now acceptable for publication, you may indicate that here to bypass the “Comments to the Author” section, enter your conflict of interest statement in the “Confidential to Editor” section, and submit your "Accept" recommendation.

Reviewer #1: (No Response)

2. Is the manuscript technically sound, and do the data support the conclusions?

Reviewer #1: (No Response)

3. Has the statistical analysis been performed appropriately and rigorously? 

Reviewer #1: (No Response)

4. Have the authors made all data underlying the findings in their manuscript fully available?

Reviewer #1: (No Response)

5. Is the manuscript presented in an intelligible fashion and written in standard English?

Reviewer #1: (No Response)

6. Review Comments to the Author

Reviewer #1: (No Response)

7. PLOS authors have the option to publish the peer review history of their article (what does this mean?). If published, this will include your full peer review and any attached files.

Reviewer #1: No

---

## [Editor Report · Acceptance letter]

23 Jan 2023

PONE-D-22-29185R1 

Serum lactate dehydrogenase is associated with impaired lung function: NHANES 2011-2012 

Dear Dr. Wei:

I'm pleased to inform you that your manuscript has been deemed suitable for publication in PLOS ONE. Congratulations! Your manuscript is now with our production department. 

Kind regards, 

on behalf of

Professor Gulali Aktas 

Academic Editor

PLOS ONE